# Circulating Biomarkers of Fibrosis Formation in Patients with Arrhythmogenic Cardiomyopathy

**DOI:** 10.3390/biomedicines11030813

**Published:** 2023-03-07

**Authors:** Stephanie M. van der Voorn, Mimount Bourfiss, Steven A. Muller, Tolga Çimen, Ardan M. Saguner, Firat Duru, Anneline S. J. M. te Riele, Carol Ann Remme, Toon A. B. van Veen

**Affiliations:** 1Department of Medical Physiology, Division Heart & Lungs, University Medical Center Utrecht, 3584 CM Utrecht, The Netherlands; 2Department of Cardiology, Division Heart & Lungs, University Medical Center Utrecht, 3508 GA Utrecht, The Netherlands; 3Department of Cardiology, University Heart Center Zurich, University Hospital Zurich, CH-8091 Zurich, Switzerland; 4Center for Integrative Human Physiology (ZIHP), University of Zurich, CH-8091 Zurich, Switzerland; 5Department of Clinical and Experimental Cardiology, Heart Centre, Amsterdam UMC, Location Academic Medical Center, University of Amsterdam, 1105 AZ Amsterdam, The Netherlands

**Keywords:** biomarkers, fibrosis, arrhythmogenic cardiomyopathy (ACM), collagen

## Abstract

Arrhythmogenic cardiomyopathy (ACM) is a progressive inheritable disease which is characterized by a gradual fibro-(fatty) replacement of the myocardium. Visualization of diffuse and patchy fibrosis patterns is challenging using clinically applied cardiac imaging modalities (e.g., late gadolinium enhancement, LGE). During collagen synthesis and breakdown, carboxy–peptides are released into the bloodstream, specifically procollagen type-I carboxy-terminal propeptides (PICP) and collagen type-I carboxy-terminal telopeptides (ICTP). We collected the serum and EDTA blood samples and clinical data of 45 ACM patients (age 50.11 ± 15.53 years, 44% female), divided into 35 diagnosed ACM patients with a 2010 ARVC Task Force Criteria score (TFC) ≥ 4, and 10 preclinical variant carriers with a TFC < 4. PICP levels were measured using an enzyme-linked immune sorbent assay and ICTP levels with a radio immunoassay. Increased PICP/ICTP ratios suggest a higher collagen deposition. We found significantly higher PICP and PICP/ICTP levels in diagnosed patients compared to preclinical variant carriers (*p* < 0.036 and *p* < 0.027). A moderate negative correlation existed between right ventricular ejection fractions (RVEF) and the PICP/ICTP ratio (*r* = −0.46, *p* = 0.06). In addition, significant correlations with left ventricular function (LVEF *r* = −0.53, *p* = 0.03 and end-systolic volume *r* = 0.63, *p* = 0.02) were found. These findings indicate impaired contractile performance due to pro-fibrotic remodeling. Follow-up studies including a larger number of patients should be performed to substantiate our findings and the validity of those levels as potential promising biomarkers in ACM.

## 1. Introduction

Arrhythmogenic cardiomyopathy (ACM), most prominently known for its subset form, namely arrhythmogenic right ventricular cardiomyopathy (ARVC), is a progressive inheritable cardiomyopathy which is characterized by fibro-(fatty) replacement of cardiomyocytes in the ventricles [1,2,3]. Pathogenic variants that trigger the initiation of the disease are mainly found in genes encoding desmosomal proteins, such as *plakophilin-2* (*PKP2*), *plakoglobin* (*JUP*), *desmoplakin* (*DSP*), *desmoglein-2* (*DSG2*) and *desmocollin-2* (*DSC2*) [1,4], as well as non-desmosomal genes such as *phospholamban* (*PLN*) [5,6]. Depending on demographics, the prevalence of ACM is estimated to be around 1:1000–1:5000 [1,7]. The first clinical presentation of ACM mostly occurs in adolescence; however, presentation is age- and sex-dependent and highly variable among patients. For diagnosis, the 2010 ARVC Task Force Criteria (TFC) score is used. This TFC score consists of six categories: depolarization and repolarization abnormalities, genetics, arrhythmias, tissue characterization and structural alterations. Clinical manifestations include premature ventricular contractions (PVCs), T wave inversions, terminal activation delays (TAD) or epsilon waves, non-sustained or sustained ventricular tachycardias (VT) or ventricular fibrillation, and fibrosis formation within both ventricles, but also in the atria [1,4,7,8].

Fibrosis formation is one of the hallmarks of ACM and is suggested to be one of the main factors involved in the disturbance of impulse propagation and the predisposition to ventricular arrhythmias (VA) and in the progression of contractile dysfunction [1]. Embedded in routine clinical care of patients, invasive and non-invasive techniques are used to monitor cardiac fibrosis. One such an invasive technique, collecting a cardiac biopsy, is not often used anymore due to its inherent risk of inadvertent perforation of the ventricular walls and the fact that the septum is mostly spared from pathological remodeling during ACM disease progression, often yielding false-negative results [3]. A non-invasive technique used in the clinic is late gadolinium enhanced cardiac magnetic resonance imaging (LGE-CMR). Administration of a gadolinium-based contrast agent visualizes fibrous tissue in the heart. However, drawbacks of this technique includes that fibrosis is visualized indirectly and only larger patches of fibrosis can be detected [9]. Therefore, T1-mapping has recently been suggested as a new technique to detect diffuse fibrosis. However, this technique is not widely used and not well validated in ACM, necessitating the need for novel, non-invasive assessments of cardiac fibrosis [10,11].

In the heart, collagen type I (85%) and collagen type III (11%) are the main types of collagen found [12]. In the human heart, the half-life of collagens is around 90–120 days. Fibroblasts synthesize these collagen types as pre-procollagen. In the endoplasmic reticulum, procollagen is formed and transported to the extracellular matrix (ECM), where the amino (N)-propeptides and carboxy (C)-propeptides are cleaved off in a 1:1:1 stoichiometry [13]. After cleavage, these propeptides are released into circulation. During the process of collagen degradation, matrix metalloproteinases (MMPs) cleave the components of the ECM into fragments, releasing collagen type-I carboxy-terminal telopeptides (ICTP) into the bloodstream [9,13]. The aim of this study was to investigate if these procollagen type-I carboxy-terminal propeptides (PICP) and ICTP might be useful as biomarkers indicative for fibrosis formation and progression of the cardiomyopathy in ACM patients. In this study, we measured PICP and ICTP levels in blood samples obtained from ACM patients and correlated these levels to clinical parameters to explore their potential as biomarkers of disease stage and fibrosis in relation to clinical outcomes.

## 2. Materials and Methods

### 2.1. Study Population

For this study, venous blood samples (EDTA and serum) of 45 ACM patients were included from Switzerland (*n* = 19) and the Netherlands (*n* = 26) after collecting signed informed written consent letters (UNRAVEL METC 12-387 and KEK-ZH-Nr. PB 2016-02109). Patients with a history of a heart transplantation (1) or atrial fibrillation (1) were excluded from the study. Of these 45 ACM patients, 35 patients fulfilled the 2010 ARVC TFC score ≥ 4 and were defined as affected (“definite”) patients. Ten patients did not fulfill those criteria (TFC score 2 or 3) and were defined as preclinical variant carriers (including “possible” and “borderline” cases). This study is part of the UCC-UNRAVEL biobank [14], a single-center research data platform that combines routine electronic health records enriched with deep phenotyping, genetic data and standardized biobanking to facilitate research on (inherited) heart diseases (www.unravelrdp.nl). The study was approved by the local institutional ethics review board (University Medical Center Utrecht, protocol UCC-UNRAVEL #12-387) and conducted according to the Declaration of Helsinki. Blood samples were prepared, aliquoted and stored (AUMC-Durrer center Biobank number DC17-006 and Zurich Biobank for ARVC and ACM Overlapping Syndromes) at −80 °C until use.

### 2.2. Clinical Data Collection

The date of blood collection was considered baseline (T = 0) in our study. Clinical data were extracted maximally two years prior or following T = 0 from the Netherlands ACM registry or Zurich ACM Registry. This two-year time window was previously validated in a cohort of *PLN* pathogenic variant carriers [15]. The Netherlands and Zurich ACM registries collect clinical information from Dutch and Swiss ACM patients and their relatives (www.acmregistry.nl and www.arvc.ch). These data include demographics, symptoms, medication use, genetic analysis and test results from electrocardiograms (ECG), Holter recordings, cardiac imaging and interventions such as an implantable cardioverter-defibrillator (ICD) and catheter ablation or development of heart failure (HF).

A proband (index patient) was defined as the first affected family member seeking medical attention for ACM-related complaints in whom diagnosis of ACM was confirmed. With MRIs, right and left ventricular ejection fraction (RVEF, LVEF) were measured. End-diastolic and end-systolic volume (EDVi, ESVi) were measured in mL indexed by body surface area (BSA). With echocardiography, RV dilatation was assessed as a qualitative assessment of the RV, which was compared to the diameter of the LV. QRS duration (ms) and T-wave inversions were assessed on 12-lead ECG recordings. A T-wave was considered inverted if the amplitude was ≥0.1 mV below baseline. A TAD was defined as the longest value in the leads V1-V3 from the nadir of the S wave to the end of all depolarization deflections in the absence of a complete right bundle branch block (BBB), and was considered prolonged if ≥55 ms. A BBB is defined as an atypical complete right BBB, defined as >120 ms + R’ in V1 but not fulfilling the typical complete right BBB (>120 ms + R’ in V1 or V2 + S > 40 ms in lead I or V5/V6) or unspecific intraventricular conduction delay [16].

### 2.3. Measurement of Fibrosis Markers

PICP levels were measured with a commercially available enzyme-linked immune sorbent assay (ELISA) (Microvue Bone CICP, Quidel, product number 8005) according to the manufacturer’s protocol. Briefly, EDTA serum was diluted 1:12 in an assay buffer and added into the wells coated with murine monoclonal anti-CICP. After incubation, anti-CICP was added to each well. Next, the enzyme conjugate, substrate solution and lastly, the stop solution were administered to each well. Optical density was measured at 405 nm using the Bio-plex^®^ 200 system (Bio-Rad, Hercules, CA, USA).

ICTP levels were measured with a radioimmunoassay (RIA) (UniQ^®^, Orion Diagnostica, product number 68601). After the serum was pipetted into the tubes, the tracer and antiserum were added. Following mixing and incubation, procollagen separation reagent was added to the tubes. Subsequently, the tubes were centrifuged, decanted and counted on a gamma counter (Wizard^2®^, PerkinElmer, Waltham, MA, USA). PICP and ICTP levels are shown as ng/mL. The PICP/ICTP ratio is used to indicate the balance between the synthesis and breakdown of fibrosis [15].

### 2.4. Statistics

Biomarker data were normally distributed, so parametric tests were used for analysis. Continuous variables between two groups were analyzed with an unpaired Student’s *t*-test, and for multiple testing, Bonferroni correction was applied. Correlations between two variables were analyzed using a Pearson correlation coefficient. A correlation was considered weak between 0.1 to 0.4, moderate between 0.4 to 0.6, strong between 0.6 to 0.8 and very strong between 0.8 to 1. A receiver operating characteristic (ROC) curve was performed to determine the diagnostic value of the biomarkers, together with the area under the curve (AUC) and confidence intervals. Data were considered significant if *p* < 0.05. Data are shown as averages ± standard deviation, except for the TFC score, which is shown as the median (interquartile range). Statistical analysis was performed using PRISM 9.4 (GraphPad Software, La Jolla, CA, USA).

## 3. Results

### 3.1. Patient Characteristics

Table 1 summarizes the patient characteristics. A total of 45 ACM patients were included in this study, of which 35 were defined as affected ACM patients (78%). The average age was 50.1 ± 15.5, 20/45 (44%) patients were female and 31/43 (72%) were probands. The median TFC score was 4 (4–7). A total of 54% (26/48) of the patients carried a heterozygous *PKP2* pathogenic variant, and six patients a *DSP* variant classified as (likely) pathogenic, variants of uncertain significance (VUS) or (likely) benign. Two carried a *JUP* variant (classified as VUS), four a *DSG2* (likely) pathogenic variant, two a *DSC2* variant likely pathogenic or benign and eight patients were gene elusive. Two patients that carried a *DSG2* variant also carried a *PKP2* pathogenic variant. In addition, one of the patients that carried a *DSP* variant also carried a *PKP2* variant.

Medication use did not differ between the preclinical variant carriers and affected ACM patients. A total of 54% of affected patients in which LGE was performed showed LGE in the RV (7/13), while in preclinical patients, no LGE in the RV was observed (0/4). RVEF was decreased, though not significant, in affected patients (41% ± 10.55) compared to preclinical variant carriers (56% ± 4.4, *p* = 0.35). In addition, in affected patients, RV EDVi (101.5 ± 25.2 mL/m^2^) did not significantly differ from RV EDVi of preclinical variant carriers (78.4 ± 11.5 mL/m^2^, *p* > 0.99). Furthermore, no differences in LVEF were found between affected (57% ± 10.73) and preclinical variant carriers (59% ± 5.19, *p* > 0.99). Finally, no differences were found in RV dilatation (*p* = 0.66) or ECG parameters, such as QRS duration, T wave inversions, TAD and BBB between preclinical and affected ACM patients (*p* > 0.99).

### 3.2. Fibrosis Biomarkers

The average level of measured PICP in all ACM patients (preclinical and affected) was 165.2 ± 61.4 ng/mL. For ICTP, the average level in all patients was 4.0 ± 1.5 ng/mL. Total collagen turnover (PICP/ICTP ratio) was 43.8 ± 17.9. No difference in PICP levels and overall collagen turnover between males and females was observed, as depicted in Figure 1A,B. When comparing ACM patients classified as preclinical variant carriers or affected patients, we found a significant difference in fibrosis biomarker levels of PICP (175.4 ± 63.8 ng/mL for affected patients compared to preclinical variant carriers 129.5 ± 35.1 ng/mL, *p* = 0.036). In addition, the PICP/ICTP ratio was significantly higher in affected patients (47.0 ± 18.4) compared to preclinical variant carriers (32.9 ± 10.8, *p* = 0.027). No difference was found in ICTP levels between the two groups. Data are summarized in Table 2.

A weak, but significant, positive correlation was found between PICP levels and TFC score *(r* = 0.31, *p* = 0.038), whereas no correlation was seen between PICP/ICTP levels and TFC score (*r* = 0.18, *p* = 0.24, Figure 1C,D). Age displayed a weak negative correlation, though not significant, with PICP levels (*r* = −0.20, *p* > 0.05), while no correlation between age and PICP/ICTP levels was observed (Appendix A). Medication use of betablockers, antiarrhythmics or angiotensin-converting enzyme (ACE)-inhibitors did not influence biomarker levels in ACM patients (Appendix A). However, patients that used diuretics had significantly higher PICP levels (236.4 ± 89.3 ng/mL) compared to preclinical variant carriers (158.2 ± 47.9 ng/mL, *p* = 0.008), as shown in Appendix A. On the contrary, diuretics had no effect on overall collagen turnover (51.6 ± 22.4 compared to 42.6 ± 17.6, *p* = 0.38, Appendix A).

### 3.3. Structural Parameters and Fibrosis Biomarkers

A significant negative correlation between LVEF and total collagen turnover was observed (*r* = −0.53, *p* = 0.03, Figure 2A). In addition, a moderate negative (and borderline significant) correlation was observed between PICP/ICTP ratios and RVEF (*r* = −0.47, *p* = 0.06, Figure 2B). Moreover, a strong significant correlation with LV ESVi (*r* = 0.63, *p* = 0.02, Figure 2C) was found, and a weak but not significant correlation with RV ESVi (*r* = 0.38, *p* = 0.21, Figure 2D) was observed. Similarly, no correlations between PICP/ICTP ratios and LV EDVi (*r* = 0.22, *p* = 0.46) nor between RV EDVi and PICP/ICTP ratios (*r* = 0.01, *p* = 0.98) were found (see Figure 2E,F). Though the vast majority of ACM patients presented with RV dilatation, this phenomenon was not associated with total collagen turnover, and the presence of RV dilatation was found in ACM patients (Figure 2G). Finally, LGE-CMR was performed only on a minor subset (*n* = 17) of ACM patients included in this study. Acknowledging the limited power of this analysis, for both RV and LV, no correlation between fibrosis biomarkers and LGE was found (Appendix A).

### 3.4. Electrical Parameters and Fibrosis Biomarkers

Fibrosis formation impacts several modes of cardiac function and may hamper electrical impulse propagation throughout the heart. No correlation was found between QRS duration (ventricular activation) and an increased PICP/ICTP ratio (Figure 3A). In addition, the presence of a T-wave inversion in at least one of the precordial leads did not correlate with elevated PICP/ICTP ratios (Figure 3B). Furthermore, seven patients showed a TAD, but no relation with total collagen turnover was found (Figure 3C). Finally, a BBB was observed in five patients; however, this was not associated with a significantly higher PICP/ICTP ratio compared to patients that did not present with a BBB (Figure 3D).

### 3.5. Diagnostic Value of Fibrosis Biomarkers

To evaluate the diagnostic value of fibrosis biomarkers for predicting a clinical diagnosis of ACM according to the 2010 TFC scores, a ROC analysis was performed. In this way, we compared the biomarker levels of the preclinical variant carriers with the levels in affected ACM patients. For PICP levels, the optimal determined cutoff value was 134.5 ng/mL with 74% sensitivity and 60% specificity, while the AUC was 0.73 ([0.57–0.89], *p* = 0.027). For PICP/ICTP, the optimal cutoff value was 32.7 with 80% sensitivity and 60% specificity, and the AUC was 0.74 ([0.58–0.90], *p* = 0.024, see Figure 4). Next, we compared clinical characteristics of the groups with low or high fibrosis biomarker levels using the determined cutoff value of the AUC. For PICP, we found significant differences in proband status, TFC score and T-wave inversion. In addition, for the PICP/ICTP ratio, significant differences were found in TFC score and RV dilatation between the two groups. No differences in other clinical characteristics such as age, sex, medication use, LGE, EF, QRS duration, TAD or BBB were found for the two fibrosis biomarkers.

## 4. Discussion

In this study, we explored the potential relation between circulating biomarkers of fibrosis and clinical disease severity in patients diagnosed with, or predisposed to, ACM. We found significantly higher PICP plasma levels and total collagen turnover in affected patients compared to preclinical variant carriers. In addition, moderate to strong correlations were found with LVEF, RVEF and LV ESVi, which may suggest that pro-fibrotic remodeling hampers the contractile function and relaxation of both ventricles. Finally, ROC analysis showed the association of fibrosis biomarkers for the diagnosis of ACM in our study.

PICP and ICTP are not markers that are specifically released from cardiac tissue but reflect the overall collagen turnover throughout the body. However, it was previously shown that PICP and ICTP levels are different in controls compared to patients with cardiac abnormalities [17,18,19]. Studies in patients with (diastolic) HF showed increased PICP levels in patients compared to individuals without this clinical phenotype [17,18]. Similar observations were made in a study of patients with diastolic dysfunction, where the latter showed significantly elevated PICP levels [20]. In a study by Rojek et al., it was reported that patients suffering from left ventricular hypertrophy carrying a particular polymorphism in the *peroxisome proliferator-activated receptor-γ coactivator-1 α* (*PPARGC1A*) gene had a higher ratio of PICP to procollagen type-III amino-terminal propeptides (PIIINP), a marker to study collagen type III [19]. These results are comparable to our findings, as we showed not only significantly higher levels of PICP, but also significant elevation of total collagen turnover in affected ACM patients compared to preclinical variant carriers. On the contrary, some studies reported comparable PICP levels in controls and patients suffering from HF [21], dilated cardiomyopathy (DCM) [22] or hypertrophic cardiomyopathy (HCM) [23].

In our study we found a weak negative, however not significant, correlation between age and PICP levels. This correlation coefficient is similar to what was found in a small study in elderly people with diastolic dysfunction [20]. Conversely, a study in HCM patients showed a weak positive correlation between PICP levels and age, implying that if a correlation with age exists, it will only have a weak effect [24]. Therefore, it is plausible to hypothesize that the increased synthesis and total collagen levels, as found in affected patients, are indeed related to increased fibrotic activity. In addition, we found significant increased PICP levels in patients using diuretics. This is in contrast to a study from López et al., where they studied the effect of two types of diuretics, torasemide (also known as torsemide) and furosemide, on fibrotic content and fibrosis biomarker levels. It was revealed that the use of torasemide decreased myocardial fibrotic content in the heart, which correlated to reduced PICP levels in these patients. On the contrary, this effect was not shown in patients using furosemide [25]. Among other types of diuretics, only two of our patients included in our study used torasemide and one patient used furosemide. However, PICP levels of one of these torasemide-using patients were high (232.1 ng/mL respectively), while the other patient showed the lowest level of PICP (97.7 ng/mL) (see marked patients in Appendix A). This might be explained by the fact that these patients using diuretics are severely affected by the disease, with the highest TFC scores and dysfunctional LV and RV. Another factor that could be of influence is the time period in which the diuretics have been used (data not available).

Recently, the prognostic value of PICP levels with LGE-CMR was tested in a population of idiopathic DCM patients. In contrast to our findings, this study by Raafs et al. found significantly higher PICP levels in patients with a positive LGE compared to LGE-negative patients [26]. Interestingly, PICP levels correlated significantly to collagen measured in a cardiac biopsy, which was also found in a study of HCM patients [24,26]. When only patients with severe HF were studied to analyze PICP and collagen content in the biopsy, the correlation became even stronger, indicating that PICP levels accurately mimic fibrotic content of the heart. Lastly, they found that patients with elevated PICP levels and positive LGE-CMR correlated to worse clinical outcome such as all-cause death, life-threatening arrhythmia or HF hospitalization [26]. These results showed that PICP levels are correlated to cardiac fibrosis and might have potential for risk stratification, as elevated PICP levels predicated outcomes in these idiopathic DCM patients. The main difference with our study includes the study setup, in which data were not simultaneously collected in a retrospective manner. Therefore, it is worthwhile to propose a larger study in which LGE and blood samples are measured simultaneously in more patients than were included in our study to examine whether this phenomenon will also be observed in an ACM patient population, with the likelihood that this association in our study was underpowered. In addition, a follow-up period of at least three years should be included to study the prognostic power of PICP and total collagen turnover for adverse events or progression of disease.

In contrast to our study, most of the previous studies have focused on either PICP or ICTP levels separately. Only a few studies have been performed where total collagen turnover was compared in patients suffering from cardiac abnormalities. In a cohort as part of the Cardiovascular Health Study, significant differences were found in the PICP/ICTP ratio between patients with HF and controls; however, in this case controls were found to have higher total collagen turnover compared to patients [21]. In a cohort of HCM patients, the ratio was correlated to clinical outcomes; however, the ratio was calculated as PICP minus ICTP instead of PICP divided by ICTP [23].

Several studies have attempted to correlate clinical outcomes to fibrosis levels in HF-diagnosed patients and those with other forms of cardiomyopathy. When fibrosis markers were correlated to echocardiography variables in HF patients, such as LVEF and interventricular septum diastolic thickness, weak but significant correlations were found [21]. In addition, in hypertensive patients, weak but significant correlations were found with echocardiographic parameters, such as estimated pulmonary capillary wedge pressure [27]. So, similar to this study, correlation coefficients were rather weak.

## 5. Conclusions

In conclusion, in this study we explored the potential relation of circulating biomarkers of fibrosis and clinical disease severity in ACM patients. We found increased ratios of PICP/ICTP in affected ACM patients showed moderate to strong correlations with ventricular contractile function, indicating hampering of contractions, potentially due to fibro-(fatty) infiltration. However only a limited subset of included patients could be analyzed. To get more substantial insight into the potential relation of fibrosis biomarkers and clinical outcomes in ACM patients, more patients should be included in a follow-up study.

## 6. Limitations

Limitations of this study include that blood sampling and clinical data were not collected on the same day, although an effort was made to take the closest examinations. In a previous study this limitation did not influence study outcomes [15]. In addition, whether patients suffered from other conditions related to fibrotic replacement unrelated to fibrosis in the heart (e.g., females after menopause and bone diseases) was unknown. These factors potentially could have influenced the outcome of this study. Lastly, only a subset of patients could be included per clinical analysis, as useful clinical data were not available for all, which resulted in groups being rather small. Although we found weak to moderate correlations, some did not reach significance, probably due to the low number of patients that could be included in the analysis. Although not considered as a real limitation, we did not implement healthy control subjects due to the fact that clinical records are obviously lacking, and therefore, it is not possible to make correlations with disease severity.

## Figures and Tables

**Figure 1 biomedicines-11-00813-f001:**
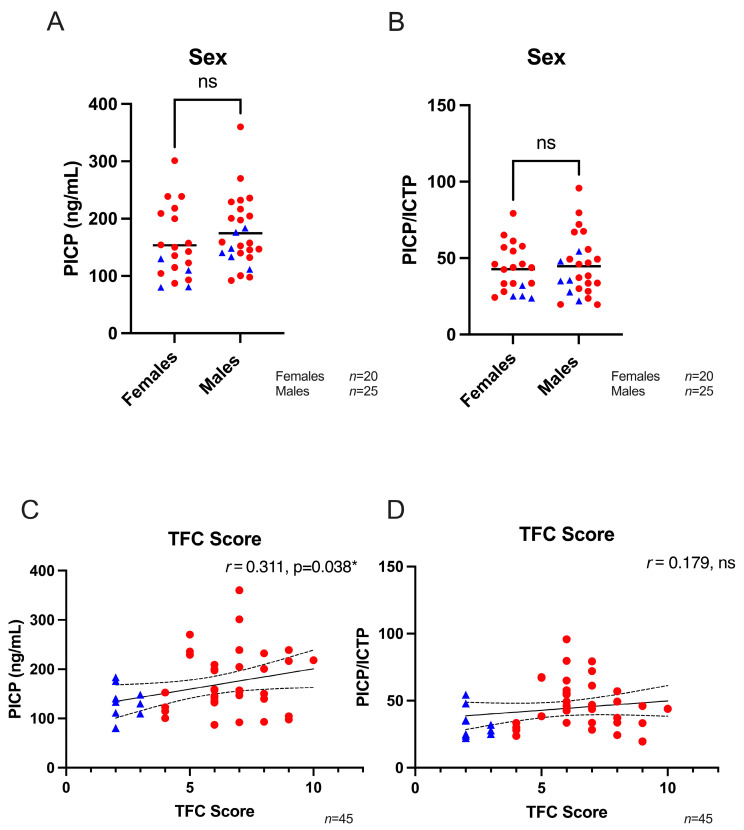
The effect of sex and TFC score on fibrosis biomarker levels. (**A**,**B**) PICP and the PICP/ICTP ratio was similar between females (*n* = 20) and males (*n* = 25). (**C**) A weak but significant correlation was found between TFC score and PICP levels (*n* = 45). (**D**) A weak correlation existed between TFC score and total collagen turnover (*n* = 45). Blue triangles represent preclinical variant carriers, while red dots are affected ACM patients. PICP; procollagen type I carboxy-terminal pro-peptide, ICTP; C-terminal telopeptide collagen type I, TFC; 2010 Task Force Criteria, *r*; Pearson correlation coefficient, ns; not significant. Unpaired Student’s *t*-test and Pearson’s correlation coefficient are performed. * *p* < 0.05.

**Figure 2 biomedicines-11-00813-f002:**
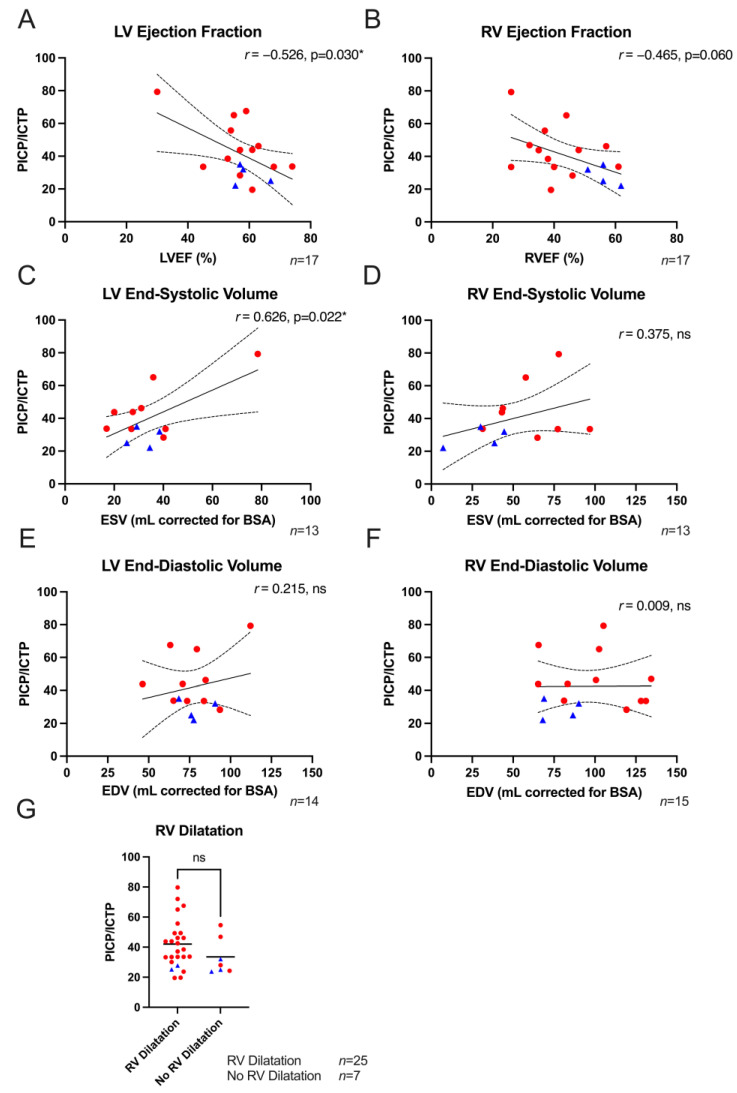
Structural parameters in relation to fibrosis biomarkers. (**A**) A significant negative correlation with LVEF (*n* = 17) was found. (**B**) A moderate negative correlation was found between total collagen turnover and RVEF (*n* = 17). (**C**) A significant strong correlation with LV ESVi (*n* = 13) was found. (**D**) A weak, although not significant, positive correlation with RV ESVi existed (*n* = 16). (**E**,**F**) A weak non-significant correlation with LV EDVi (*n* = 14) existed, while no correlation was found with RV EDVi (*n* = 18). (**G**) PICP/ICTP ratios were similar in patients with RV dilatation (*n* = 25) and patients with normal RV dimensions (*n* = 7). Blue triangles represent preclinical variant carriers, while red dots are affected ACM patients. PICP; procollagen type I carboxy-terminal pro-peptide, ICTP; C-terminal telopeptide collagen type I, RV; right ventricle, LV; left ventricle, EF; ejection fraction, EDV; end-diastolic volume, ESV; end-systolic volume, BSA; body surface area, *r*; Pearson correlation coefficient, ns; not significant. Unpaired Student’s *t*-test was performed, and Pearson’s correlation coefficient was assessed. * *p* < 0.05.

**Figure 3 biomedicines-11-00813-f003:**
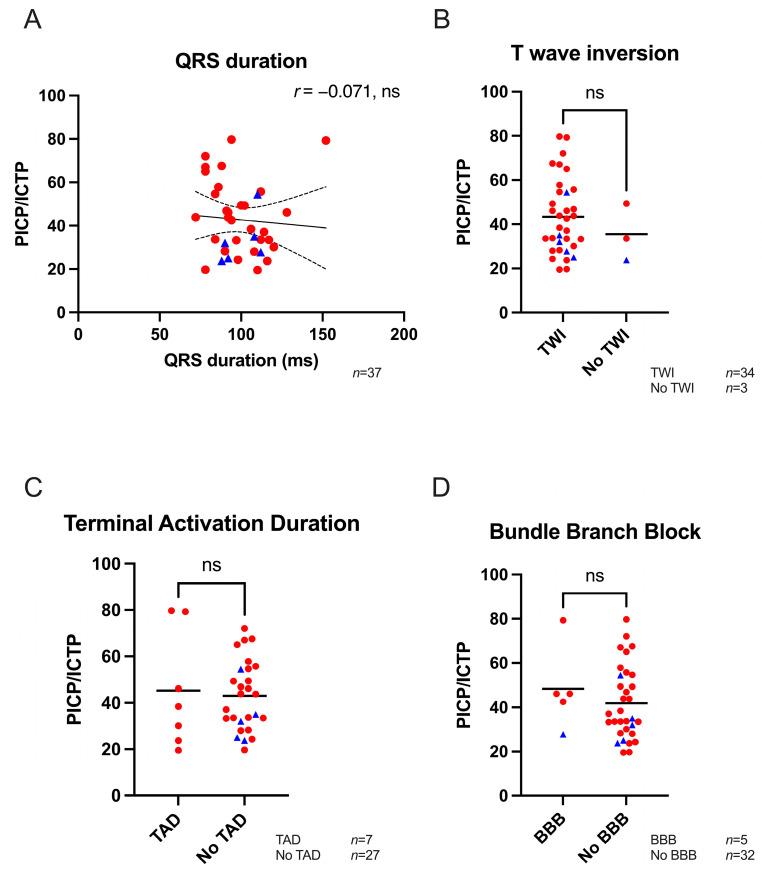
Electrical parameters and total collagen turnover biomarkers. (**A**) No correlation was found between PICP/ICTP ratio and QRS duration (*n* = 37). (**B**) Similar PICP/ICTP ratios were found in patients with (*n* = 34) or without (*n* = 3) a T-wave inversion. (**C**,**D**) No difference in PICP/ICTP ratios were found in patients with a TAD (*n* = 7) or BBB (*n* = 5) when compared to patients without the presence of a TAD (*n* = 27) or BBB (*n* = 32). Blue triangles represent preclinical variant carriers, while red dots are affected ACM patients. PICP; procollagen type I carboxy-terminal pro-peptide, ICTP; C-terminal telopeptide collagen type I, TWI; T wave inversion, TAD; terminal activation duration, BBB; bundle branch block, *r*; Pearson correlation coefficient ns; not significant. Unpaired Student’s *t*-test was performed, and Pearson’s correlation coefficient was assessed.

**Figure 4 biomedicines-11-00813-f004:**
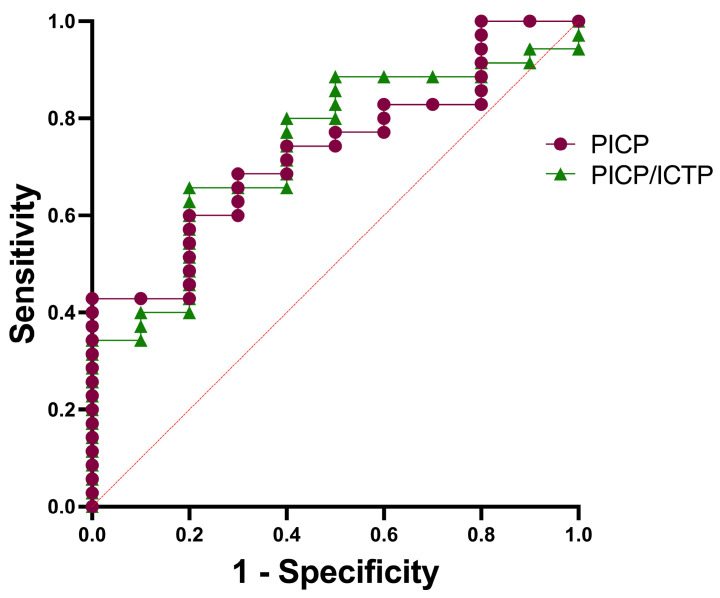
Predicting ACM according to ROC curves of PICP and PICP/ICTP levels. Area under the curve (AUC) of PICP is 0.73 [0.57–0.89]. AUC of PICP/ICTP is 0.74 [0.58–0.90]. PICP; procollagen type I carboxy-terminal pro-peptide, ICTP; C-terminal telopeptide collagen type I.

**Table 1 biomedicines-11-00813-t001:** Patient characteristics of the 45 ACM patients included in this study. The patients were divided into preclinical variant carriers (*n* = 10) and affected (*n* = 35) patients. ACM; arrhythmogenic cardiomyopathy, TFC; 2010 Task Force Criteria, *PKP2*; *plakophilin-2*, *DSP*; *desmoplakin*, *JUP*; *plakoglobin*, *DSG2*; *desmoglein-2*, *DSC2*; *desmocollin-2*, ACE; angiotensin-converting enzyme, CMR; cardiac magnetic resonance imaging, LGE; late gadolinium enhancement, RV; right ventricle, EF; ejection fraction, EDV; end-diastolic volume, ESV; end-systolic volume, LV; left ventricle, ECG; electrocardiogram, TAD; terminal activation duration, BBB; bundle branch block.

	All ACM (*n* = 45)	Preclinical Variant Carriers (*n* = 10)	Affected Patients (*n* = 35)	Adjusted *p*-Value
**Demographics**				
Age (years)	50.11 ± 15.53	51 ± 17.41	49.86 ± 15.22	>0.999
Female	20/45 (44)	4/10 (40)	16/35 (46)	>0.999
Proband Status	31/43 (72)	3/10 (30)	28/33 (85)	0.011 *
TFC Score	4 (4–7)	2 (2–2.75)	7 (6–8)	<0.003 **
**Genetic variant**				
*PKP2*	26/48 (54)	5/10 (50)	21/38 (55)	>0.999
*DSP*	6/48 (13)	1/10 (1)	5/38 (13)	>0.999
*JUP*	2/48 (4)	2/10 (20)	0/38 (0)	>0.999
*DSG2*	4/48 (8)	0/10 (0)	4/38 (11)	>0.999
*DSC2*	2/48 (4)	0/10 (0)	2/38 (5)	>0.999
No Variant Detected	8/48 (17)	2/10 (20)	6/38 (16)	>0.999
**Medication**				
Betablockers	17/39 (44)	2/7 (29)	15/32 (47)	>0.999
Antiarrhythmics	19/38 (50)	2/6 (33)	17/32 (53)	>0.999
ACE-inhibitors	5/38 (13)	0/6 (0)	5/32 (16)	>0.999
Diuretics	6/38 (16)	0/6 (0)	6/32 (19)	>0.999
**Imaging/CMR**				
LGE RV	7/17 (41)	0/4 (0)	7/13 (54)	>0.999
RVEF (%)	44.23 ± 11.54	56.21 ± 4.43	40.96 ± 10.55	0.351
RV EDVi (mL/m^2^)	95.35 ± 24.36	78.42 ± 11.52	101.50 ± 25.19	>0.999
RV ESVi (mL/m^2^)	50.50 ± 23.92	30.12 ± 16.40	59.56 ± 21.39	0.899
LGE LV	6/16 (38)	2/3 (67)	4/13 (31)	>0.999
LVEF (%)	57.44 ± 9.44	59.37 ± 5.19	56.69 ± 10.73	>0.999
LV EDVi (mL/m^2^)	77.55 ± 15.70	78.18 ± 9.23	77.30 ± 18.10	>0.999
LV ESVi (mL/m^2^)	34.25 ± 15.21	31.83 ± 5.85	35.33 ± 18.16	>0.999
**Echocardiogram**				
RV Dilatation	25/32 (78)	2/5 (40)	23/27 (85)	0.662
**ECG parameters**				
QRS duration (ms)	99.54 ± 16.53	100 ± 11.1	99.45 ± 17.53	>0.999
T Wave Inversion	34/37 (92)	5/6 (83)	29/31 (94)	>0.999
TAD ≥ 55 ms	7/34 (21)	0/5 (0)	7/29 (24)	>0.999
BBB	5/37 (14)	1/6 (17)	4/31 (13)	>0.999

Unpaired Student’s *t*-test is performed. ** *p* < 0.01, * *p* < 0.05.

**Table 2 biomedicines-11-00813-t002:** Levels of fibrosis biomarkers in all ACM patients (*n* = 45) or divided into preclinical variant carriers (*n* = 10) and affected (*n* = 35) ACM patients. ACM; arrhythmogenic cardiomyopathy, PICP; procollagen type I carboxy-terminal pro-peptide, ICTP; C-terminal telopeptide collagen type I.

	All ACM (*n* = 45)	Preclinical Variant Carriers (*n* = 10)	Affected Patients (*n* = 35)	*p*-Value
PICP (ng/mL)	165.18 ± 61.43	129.47 ± 35.11	175.38 ± 63.84	0.036 *
ICTP (ng/mL)	4.02 ± 1.46	4.02 ± 0.71	4.02 ± 1.62	0.991
PICP/ICTP	43.84 ± 17.91	32.89 ± 10.79	46.97 ± 18.40	0.027 *

Unpaired Student’s *t*-test is performed. * *p* < 0.05.

## Data Availability

The data presented in this study are available on request from the corresponding author.

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
