# Peer review of "Circulating Biomarkers of Fibrosis Formation in Patients with Arrhythmogenic Cardiomyopathy"

_biomedicines, 2023, doi:10.3390/biomedicines11030813_

Round 1

Reviewer 1 Report

Thank you very much for the opportunity to read an interesting paper on Biomarkers of Fibrosis Formation in Patients with Arrhythmogenic Cardiomyopathy. The disease covered by the publication is still of great interest to cardiologists. Although its incidence is fortunately relatively rare, the consequences are unfortunately often very serious, leading to life-threatening arrhythmias and even sudden cardiac death in a group of young patients.

The presented work is interesting; the tested markers look promising; I am convinced that this direction of research may, in the future, have a significant impact on the clinical methods of diagnosing patients for arrhythmogenic cardiomyopathy, which at present is certainly not perfect.

I have no significant comments about the work. Instead, I would like to ask for a few words of clarification on the following issues that I found interesting while reading:

1. Could we find out what group of 10 patients is, on the one hand, diagnosed with ACM but, on the other hand, does not meet the 2010 ARVC criteria? I want to ask for a few sentences describing this group - how it was selected and what kind of patients they are from a clinical perspective.

2. If I understand correctly, was the ROC analysis performed based on data from patients diagnosed with ACM? Isn't that, in a way, a limitation of this analysis? However, don't the authors think that it would be worth repeating such an analysis, also using the data of patients who have not been diagnosed with ACM - thanks to this, we can look for the cut-off point of parameters describing collagen turnover for diagnosing ACM in the general population.

3. Analyzing the supplementary data, I noticed that less than half of the study group - patients diagnosed with ACM, take b-blockers, and only a small percentage take ACE-I. In the case of such a diagnosis, we should expect higher rates of use of these drugs, especially b-blockers. May I have a brief comment on this treatment? What, in the opinion of the authors, results from it?

4. In the authors' opinion, can the observed collagen turnover, or rather differences in PICP between the defined subgroups, result from other clinical reasons than the diagnosis of ACM? Can we identify or suspect clinical factors potentially confounding this interpretation of results?

Reviewer 2 Report

Review

Circulating Biomarkers of Fibrosis Formation in Patients with Arrhythmogenic Cardiomyopathy

biomedicines-2251589

Summary:

The study analyzes a cohort of 45 ACM patients. These patients were characterized for genetic variants, medication, imaging MRI, echocardiography, and ECG. The 45 ACM patients were analyzed for their PICP and ICTP peptide levels in EDTA serum. The study systematically tested different associations of PCIP/ICTP in relation to sex, TFC score, LV function, RV function, electrical parameters, and LV/RV LGE parameters measured with MRI. The most significant association was found for PCIP serum levels and TFC score. All other associations were not significant. The authors conclude that larger cohorts are needed to validate the association of PCIP/ICTP serum levels with cardiac fibrosis in ACM patients.

Major comments:

The study of van der Voorn et al. is written in good English, well structured and with some improvements acceptable for publication.

The Reviewer suggests strongly to clarify the main message of the study. In fact, the authors show that measuring PCIP and/or ICTP is NOT a biomarker to predict cardiac fibrosis in the analyzed ACM cohort! Moreover, there is no correlation of PCIP and/or PCIP/ICTP with ACM severity. Positive results of PCIP and/or PCIP/ICTP correlation are probably cohort dependent and need to be reevaluated carefully. One of the causes for this result is the insufficient patient number!

Terminal collagen I peptides are generated under circumstances were increased levels of collagen I are produced (PICP) or degraded (ICTP). Elevated PCIP or ICTP levels typically measure increased bone turnover; the major tissue site of collagen I synthesis. Alterations in PICP or ICTP reflect bone turnover and or massive collagen I alterations in another type of tissue. Due to moderate contribution of cardiac fibrosis to overall collagen I serum levels it is unlikely that these markers are sufficiently sensitive to detect early fibrosis in ACM. In the future improved T1 mapping and LGE can be an early predictor. The work should focus on improving the T1 mapping/LGE data source. The authors should clearly discuss the technical and conclusive differences compared to the Raafs et al. DCM study. This study showed strongest correlation of LGE plus PICP with DCM severity and outcome. The cohort was much larger but of similar age/sex range. In the present ACM study the patient number is insufficient!

The authors did not consider in the ACM cohort bone diseases, menopause (20/45 patients are women) as all these factors strongly affect collagen I turnover. The correlation with age (S1) shows that PCIP, PCIP/ICTP is in fact randomly distributed for the ACM and preclinical group.

The term preclinical variant carrier needs definition. This is individuals with an L(P) variant but no phenotype? These people are no ACM patients and the inclusion in the study is misleading. The authors need to clarify if this subgroup are ACM patients. From the TFC score these patients appear unaffected.

Table 1. The group “Preclinical variant carriers” was used as control group. It is unclear how this was done and whether it is justified. These patients have signs of ACM but a low TFC score. Is the TFC score of 2 considered as unaffected or affected? Why did the authors include the “Preclinical variant carriers” subgroup into the correlation analysis? The Reviewer assumes this will negatively affect the correlation outcome.

The group “Preclinical variant carriers” is small which limits comparative evaluation. Why do the authors not use an age, sex matched control cohort?

The genetic information is not presented according to current standards. What are the genetic variants regarding pathogenic (P), likely pathogenic (LP), or variant of unknown significance (VUS)? Please consider that PKP2 truncating variants, representing the largest group in the study, are no longer considered as disease causing per se (PMID: 35536239)! The genetic information has not been tested for PCIP association. Does the genetic information contribute to the TFC score? Why do the authors present the genetic information?

Minor comments:

Table 1. Treatment only 17/39 patients. It is unclear if the remaining 6 patients did not receive a medication or were not assessed. The reviewer would suggest the term medication.

Table 1. The group “Preclinical variant carriers” was used as control group. It is unclear how this was done and whether it is justified. These patients have signs of ACM but a low TFC score. Is the TFC score of 2 considered as unaffected?

Table 1. The authors should check thru the table. Some sum numbers do not match to subgroup numbers e.g. RV dilatation and BBB.

Table 1. Unpaired T-test appears twice.

Figure 3D. The abbreviation of column in figure is incomplete “No BBB”

Why measuring PCIP and ICTP? Both parameters PCIP alone and PCIP/ICTP show the same direction. So why doing two measurements? You show in Figure 4 similar sensitivity.

Reviewer 3 Report

The article deals with an extremely interesting subject with special practical implications regarding a not very common condition from cardiovascular pathology.

It is well designed, clear, concise.

The experience of the authors in the field of this pathology can be observed.

The study clearly helps to deepen the theoretical but also practical knowledge about a severe cardiovascular pathology.

It is wellknown that fibrosis formation is one of the hallmarks of ACM, and is suggested to be one of the main factors involved in disturbance of impulse propagation and the predisposition to ventricular arrhythmias.

Without doubt this can be a step forward in the non-invasive assessment of the severity of ACM in relation to clinical outcome.

Author Response

We would like to thank the reviewer for the performed efforts to evaluate our manuscript and his/her sincere enthusiasm and positive advice regarding our manuscript.